# Placental Transfer Efficiency of Neutralizing Antibodies on SARS-CoV-2 Vaccination before and after Pregnancy in Mexican Women

**DOI:** 10.3390/ijms25031516

**Published:** 2024-01-26

**Authors:** Rebeca Martínez-Quezada, Carlos Emilio Miguel-Rodríguez, Tito Ramírez-Lozada, Omar Esteban Valencia-Ledezma, Gustavo Acosta-Altamirano

**Affiliations:** 1Unidad de Investigación, Hospital Regional de Alta Especialidad de Ixtapaluca, Carretera Federal México-Puebla Km. 34.5, Pueblo de Zoquiapan, Ixtapaluca 56530, Mexico; rebemarq_710@hotmail.com (R.M.-Q.); carlos_emilio_13@hotmail.com (C.E.M.-R.); esteban84valencia@gmail.com (O.E.V.-L.); 2Consejo Mexiquense de Ciencia y Tecnología (COMECYT), Paseo Colón N° 112-A, Ciprés, Toluca 50120, Mexico; 3Unidad de Ginecología y Obstetricia, Hospital Regional de Alta Especialidad de Ixtapaluca, Carretera Federal México-Puebla Km. 34.5, Pueblo de Zoquiapan, Ixtapaluca 56530, Mexico; titolozada@yahoo.com.mx

**Keywords:** placental transfer, immunity, IgG, neutralizing antibodies, COVID-19 vaccines

## Abstract

The protection of the neonate against pathogens depends largely on the antibodies transferred placentally from the mother; for this reason, maternal vaccination against emerging viruses, such as SARS-CoV-2, is of vital importance. Knowing some of the immunogenic factors that could alter the placental transfer of antibodies could aid in understanding the immune response and neonatal protection after maternal vaccination. In this study, we analyzed the efficiency of the placental transfer of binding and neutralizing antibodies, as well as some factors that could alter the passive immune response, such as the trimester of gestation at the time of immunization, the number of doses received by the mother and the type of vaccine. Binding IgG antibodies were detected by ELISA, and the detection of neutralizing antibodies was carried out using flow cytometry. Our results show efficient transfer rates (>1), which are higher when maternal vaccination occurs during the third trimester of gestation. Antibodies are detectable in mothers and their neonates after 12 months of maternal immunization, suggesting than the vaccination against COVID-19 before and during pregnancy in the Mexican population induces a lasting neutralizing response in mothers and their newborns.

## 1. Introduction

The diagnosis and monitoring of COVID-19 in pregnant women became especially relevant, given that evidence suggests that this disease is associated with multiple adverse perinatal outcomes [1,2,3].

Among patients with symptomatic COVID-19, pregnancy is associated with an increased risk of intensive care unit admission, need for extracorporeal membrane oxygenation, and maternal death. Furthermore, severe illness during pregnancy is associated with an increased risk of obstetric and neonatal complications, including cesarean delivery, hypertensive disorders of pregnancy, preterm birth, venous thromboembolism, neonatal intensive care unit admission, lower birth weight and, potentially, stillbirth [1,2,3].

Until May 2021, in Mexico, the maternal mortality ratio due to COVID-19 was 46.6 per 100,000 live births. At that time, this disease was the main cause of maternal death; however, towards 2023, it became the sixth cause of death [4].

Vaccination against COVID-19 for pregnant women began in May 2021, highlighting the protective role of immunization on the severity of the disease in this vulnerable group of the population.

There are several vaccine platforms against COVID-19, including those based on mRNA (for example, Pfizer/BioNTech and Moderna), those containing viral vectors (AstraZeneca, CanSino and Sputnik V), those with recombinant proteins plus adjuvants (Novavax), and those with inactivated viruses (CoronaVac). Despite their emerging use, vaccines have been shown to be safe and effective, eliciting a strong cellular and humoral immune response [5,6].

Cellular immunity plays an important role in modulating disease severity and resolving infection, while in protective immunity, neutralizing antibodies play a key role in preventing infection [5].

It is worth mentioning that vaccination not only protects the mother, but also the fetus and neonate, through the placental transfer of neutralizing antibodies, which is particularly important since it can reduce the risk of severe disease from COVID-19 during the first months of life [1,7,8].

Although some authors have confirmed the placental transfer of antibodies against SARS-CoV-2 [2,9,10,11,12], the degree of transplacental passive immunity induced by maternal vaccination is not very clear, with variable results found according to the authors [13,14,15]. Furthermore, the information available to date focuses mainly on the effectiveness of mRNA vaccines; however, a large part of the Mexican population has been immunized with viral vector vaccines.

We consider that knowing some of the immunogenic factors that could alter the efficiency of placental transfer of antibodies could help understand the immune response and neonatal protection after maternal vaccination and provide better strategies for vaccination against COVID-19 in vulnerable populations. Thus, the objective of the present study was to evaluate the efficiency of the placental transfer of neutralizing antibodies against SARS-CoV-2 in vaccinated and unvaccinated women, as well as to analyze the influence of factors such as the time of immunization (before and during pregnancy), the dose and platform of vaccination, and the time between immunization and delivery.

## 2. Results

### 2.1. Characteristics of Study Population

A total of 114 mothers and their 117 neonates were included in the study. Maternal age was 24.5 years (IQR: 19–31.25), with a gestational age of 39 weeks (IQR: 38–40).

None of the participants had COVID-19 during pregnancy or at the time of admission for delivery; a small proportion (7%) reported having had the disease before pregnancy.

The control group consisted of 26 unvaccinated women; the cases of women immunized with at least one dose of a vaccine against COVID-19 represented 77.2% of the study population (Table 1).

Regardless of maternal vaccination status, at the time of study, 98.9% of mothers and neonates had detectable binding IgG titers, while neutralizing antibodies against SARS-CoV-2 were detected in 94.9%. When adjusted for vaccination status, it was found that, among women who received at least one immunization, 98.9% of maternal and neonatal samples had detectable levels of neutralizing antibodies, while among the unvaccinated population, they were detected in 80.8%.

### 2.2. Binding IgG Antibody Response According to ELISA

Concerning binding IgG, antibody titers in maternal blood do not differ significantly (*p* = 0.0943) from those found in the neonates [1.365 (IQR: 0.54–2.15) vs. 1.497 (IQR: 0.61–2.2) OD450] (Figure 1A). The results show that the antibodies were transferred efficiently, with a mean transfer ratio equal to 1.3 (±0.75) (Figure 1B).

### 2.3. Neutralizing Antibody Response by Flow Cytometry

A positive correlation was found between maternal and neonatal nAb concentrations (rs = 0.795, *p* < 0.0001); however, no statistically significant differences were found in concentrations between both groups [250.0 (IQR: 142.8–574.5) vs. 336.3 (IQR: 137.1–570.4) µg/mL] (*p* = 0.1974). The data for each participant are shown in Figure 2A. The neutralization percentage did not show statistically significant differences (*p* = 0.3013); neutralization in maternal samples was 46.09% (IQR: 31.05–70.63), while in neonatal samples it was 54.47% (IQR: 30.09–69.80) (Figure 2B). The nAb were transferred efficiently, with a mean transfer ratio of 1.24 (±0.78) (Figure 2C).

### 2.4. Effect of the Number of Doses on Placental Transfer Efficiency

We found no statistically significant differences between the number of immunizations received by the mother and the transfer ratio (*p* = 0.9067). They were calculated at 1.37 (±1.01), 1.31 (±0.97), 1.11 (±0.61), and 1.39 (±0.15), for one, two, three, and four doses, respectively. Similarly, an effective transfer was observed in the unvaccinated population [1.15 (±0.48)] (Figure 3).

### 2.5. Effect of Gestational Age on Placental Transfer Efficiency

Of all participants, only 32 provided sufficient information to calculate the gestational age at which they were vaccinated. No significant association was observed between the week of gestation at the time of the last immunization and the placental transfer rate (r = −0.015, *p* = 0.9566) (Figure 4A). However, transfer ratios were compared in those women who were immunized before and after pregnancy. When vaccination occurred before pregnancy, nAb were transferred with a mean transfer ratio of 1.1 (±0.73); in the first, second and, third trimesters, the ratios were 1.56 (±0.71), 1.21 (±1.2), and 1.84 (±0.98), respectively. Although no statistically significant differences were found (*p* = 0.2235), the average transfer after immunization in the first and third trimester of pregnancy was considerably elevated, compared to vaccination before pregnancy (Figure 4B).

### 2.6. Association between Latency since Last Immunization and Placental Transfer Efficiency

The results do not demonstrate a significant association between latency since the last dose received and transfer efficiency at time of delivery (r = 0.1441, *p* = 0.5223). Surprisingly, after 500 days (16 months), elevated levels of nAb continued to be observed in the neonates, suggesting that antenatal protection is long-lasting (Figure 5).

## 3. Discussion

Cohort studies of pregnant women are useful to determine the prevalence of immunity, its variation over time, and the duration of protection, as well as the effectiveness of vaccination [16].

Vaccination against COVID-19 in pregnant women has been shown to induce a robust immune response, which becomes comparable to that observed in non-pregnant women of reproductive age [17].

Given that vaccination contributes to providing immunological protection, it is expected that the prevalence of anti-SARS-CoV-2 nAb is high among the vaccinated population. We found that almost all women who received at least one dose of the COVID-19 vaccine had detectable levels of nAb. Interestingly, high seropositivity was also found among unvaccinated women; 100% of unvaccinated women and their neonates had detectable titers of binding IgG antibodies, while neutralizing antibodies were detected in 80% of maternal and neonatal samples.

In reference to this last point, some studies have shown that seroprevalence in unvaccinated pregnant women has been increasing after each wave of the pandemic. In this regard, between the second and third waves, the prevalence in South Africa was close to 60%, and after the end of the third wave in The Gambia, the prevalence increased to 74.6% [16,18]. As indicated by [16], the results, in these cases, could be attributable to underestimated previous SARS-CoV-2 infections, since a good proportion of the patients are asymptomatic.

In our study, only one of the unvaccinated women reported having a previous history of COVID-19, suggesting, on the one hand, the aforementioned diagnostic underestimation or, on the other, that herd immunity has been achieved.

Regardless of vaccination status, we determined nAb concentrations in mothers and neonates, finding a positive correlation between both groups (rs = 0.795, *p* < 0.0001). Neonate nAb concentrations were slightly (non-significantly) higher than maternal concentrations (Figure 2B).

It has been shown that, for a variety of pathogens, including SARS-CoV-2, the endosomal transport of IgG via the neonatal Fc receptor (FcRn), across the trophoblastic barrier, contributes to the antibody titer in blood of the umbilical cord, which is higher than that detected in maternal blood [9]. In fact, Kugelman et al. (2022) [19] indicated that antibody titers in the neonate were 2 times higher than in their maternal counterpart.

In addition to the concentration, we calculated the percentage of neutralization to be 46.09 in the mothers and 54.47 in the neonates (Figure 2B); however, to date, no exact figures have been reported that indicate adequate correlates of protection.

As mentioned, the protection of the neonate depends on the antibodies transferred from the mother; however, although some studies have analyzed the degree of passive immunity through the placenta, induced by maternal vaccination against SARS-CoV-2 [13,14,15,20,21,22,23,24], the results have been discordant.

Some authors have reported inefficient transfer across the placenta. For example, Marshall et al. (2022) [20] found significantly lower antibody titers in neonates than in their mothers; Nir et al. (2021) [14] reported a placental transfer ratio of 0.77, while Rottenstreich et al. (2021) [15] indicated ratios of 0.44 and 0.34 (anti-S IgG and anti-RBD IgG), respectively.

Setting the effective transfer of binding IgG, Beharier et al. (2021) [13] reported that the transfer ratio exceeded that observed in pregnant women with SARS-CoV-2 infection in the third trimester; in turn, Mithal et al. (2021) [21], Zdanowski and Waśniewskii (2021) [24], and Rottenstreich et al. (2022) [23], calculated ratios of 1.0, 1.28, and 1.3, respectively. As far as the nAb are concerned, they were transferred efficiently, with ratios that reached as high as 1.9 [22,23].

In agreement, our results show an efficient placental transfer, with a mean ratio of 1.24 (±0.78) for the case of nAb (Figure 2C). Likewise, the results are comparable with those of other vaccinable viruses, such as rubella, measles, whooping cough, influenza, and hepatitis B. In these cases, transfer ratios range between 0.8 and 1.7 [15].

It has been suggested that various factors, including the number of doses administered, gestational age at time of immunization, and the vaccination platform, can significantly impact the efficacy of immunization [17,25].

Regarding the effect of the number of immunizations, a study showed that IgG levels are significantly higher in mothers and their neonates after the administration of the booster dose, compared to the primary immunization (with two doses) of Pfizer [19]. Similarly, another study revealed that the booster dose with the Pfizer and Moderna vaccines significantly increases binding IgG and nAb titers when compared to a primary vaccination scheme, both in maternal and umbilical cord blood; the antibodies induced by the booster dose were efficiently transferred across the placenta [8].

The results of the present study do not show statistically significant differences between the number of doses received by the mother and the nAb concentration or placental transfer efficiency (Figure 3).

Concerning to the gestational age at time of immunization, contrary to our findings (Figure 4A), Zdanowski and Waśniewskii (2021) [24] demonstrated that there is a positive correlation between the weeks of gestation in which the first and the second dose was administered and the respective transfer ratio.

In line with this, Rottenstreich et al. (2021) [15] indicated that the optimal time for maternal vaccination is during the third trimester of gestation and that nAb transfer is higher when the mother is vaccinated at the beginning of this trimester. Similarly, our data suggest a better transfer efficiency in said trimester since the highest ratio was observed here (data not statistically significant) (Figure 4B).

For their part, Atyeo et al. (2022) [17] found that the transfer ratio of antibodies generated by vaccination is higher in the first and second trimesters than in the third; curiously, in the third trimester, a high response of maternal functional antibodies was observed, compared to the second trimester.

Regarding the latency from last vaccination to the moment of delivery, some studies agree that the increase in placental transfer is positively correlated with the weeks that have elapsed since the last vaccination dose [21,26]. In contrast, one study reported an inverse association between time since vaccination and antibody titers [19]. Our results did not demonstrate a significant correlation (r = 0.1441, *p* = 0.5223) (Figure 5).

Despite discrepancies in the time from vaccination to delivery and transfer ratios, some studies have suggested that the immune response after the administration of the booster dose, with different vaccination platforms, is stable between 6 and 8 months [20,27,28]. Regarding this topic, we found detectable levels of nAb almost 2 years after the last immunization, and the transfer was effective in participants who were vaccinated 16 months before delivery (Figure 5).

On the other hand, it has been shown that the vaccine platform can affect the efficiency of placental transfer. A study showed that the Pfizer, Moderna, and Ad26.COV2.S (Jansen) vaccines promote effective placental transfer; however, the latter (viral vector vaccine) was shown to have lower efficiency than those derived from mRNA platforms [17].

In Mexico, there are no studies that demonstrate the impact of the type of vaccine on placental transfer; however, a study carried out in the general population indicated that vaccination with Pfizer and Moderna has the best neutralization rates when compared with the AstraZeneca, Sinovac, and CanSino vaccines [29].

The results coincide with those reported by Zhang et al. (2022) [5], who demonstrated that mRNA vaccines are more immunogenic than viral vector and recombinant protein vaccines.

Evidence suggests that there are different immunological memory profiles for different vaccination platforms. For example, neutralizing antibody titers after 6 months of immunization were higher with the Moderna and Pfizer vaccines than with the Jansen vaccine, with differential kinetics for each vaccine platform. Similarly, B cell memory, specific against the Spike protein, is greater after vaccination with Moderna and Pfizer than with Jansen. Furthermore, a distinctive feature of vaccination with the latter vaccine is the high frequency of CXCR3^+^ memory B cells, suggesting a specific functional role in the B cell response with viral vectors [5].

Furthermore, although the use of homologous and heterologous boost is effective, the response with both strategies is also different. It has been shown that the heterologous boost largely increases the titer of neutralizing antibodies with respect to the homologous one [30]. In this regard, some studies have confirmed that the use of a heterologous scheme with two doses of AstraZeneca followed by reinforcement with an mRNA vaccine is more [30,31,32] immunogenic than the homologous vaccination with AstraZeneca [30,31,32].

Although we observed a certain trend towards the greater effectiveness of homologous or heterologous immunization with the Pfizer vaccine, the size of our sample represented a limitation to performing an adequate comparative analysis. Although more than 70% of the population was vaccinated, there was great variability between the data obtained. The participants received at least one dose of the Pfizer, AstraZeneca, CanSino, or Sputnik V vaccines; the combinations were widely diverse and the number of participants in each study group generated was very small, making it impossible for us to perform an adequate analysis and obtain accurate results.

Having a larger and more homogeneous cohort would allow us to have more conclusive findings on the specific immune response of each vaccination platform and its association with each of the factors that could be affecting the transfer of neutralizing antibodies across the placenta.

It is important to mention that, as seen, most of the studies carried out to date focus on the effectiveness of the Pfizer and Moderna vaccines, and few studies have been conducted with non-mRNA vaccines. More research is necessary to obtain accurate data on the efficiency of the placental transfer of antibodies induced by viral vector vaccines, such as those that were received by a large part of our study population.

Taken together, our data demonstrate neonatal protection against SARS-CoV-2 via transplacental passive immunity in the Mexican population. The results suggest that vaccination confers immunity for a long time after the last immunization, highlighting the importance and effectiveness of vaccination. Further studies are required to investigate the factors that significantly affect the efficiency of placental transfer. Furthermore, understanding the differences in vaccine efficacy and antibody transfer between the different COVID-19 vaccine platforms could guide us in the design of better vaccination and maternal–fetal and neonatal protection strategies.

## 4. Materials and Methods

### 4.1. Study Design

A prospective, cross-sectional study approved by the Research Committee (NR-57-2022) and the Research Ethics Committee (NR-CEI-HRAE-11-2022) of the Regional High Specialty Hospital of Ixtapaluca was carried out. The patients were recruited in 2 hospital care centers in Estado de México, one secondary and another tertiary care, during the period from September 2022 to March 2023. Eligible participants included all pregnant women admitted to the Gynecology and Obstetrics Unit for delivery who were willing to participate and able to give informed consent, and had a complete, partial, or null COVID-19 vaccination scheme. Participants completed a survey about the characteristics of their pregnancy, SARS-CoV-2 infection, or presence of COVID-19 symptoms before and during pregnancy and at the time of admission, as well as COVID-19 vaccination data. The women reported being immunized with one of the following vaccines in a homologous or heterologous scheme: Pfizer (BNT162b), AstraZeneca (AZD1222), Cansino (Ad5-nCoV-S), and Gam-COVID-5-Vac (Sputnik V). A total of 114 women were enrolled in the study, of which 111 had singleton pregnancies and 3 had twin pregnancies.

### 4.2. Samples Collection

After written informed consent, maternal blood samples were collected during admission and umbilical cord blood (fetal placental side) immediately after delivery. Maternal blood was collected by venipuncture in Vacutainer tubes. After cutting the umbilical cord, a clamp was used to stop the bleeding, the cord was cleaned, and the blood was then drawn directly from the vein and transferred into Vacutainer tubes.

The samples were stored at −20 °C until use.

### 4.3. Detection of Anti-SARS-CoV-2 IgG Antibodies (Spike Trimer) with ELISA

IgG antibody titers were analyzed with an Anti-SARS-CoV-2 Antibody IgG Titer Serologic Assay Kit (Spike Trimer) (Acro BioSystems, San Diego, CA, USA, Cat # RAS-T025) according to the manufacturer’s instructions. This kit is used to measure anti-SARS-CoV-2 IgG antibody titers using an indirect ELISA. The SARS-CoV-2 Spike protein was immobilized on the plate. Next, the samples are added and incubated and the wells were subsequently washed. The secondary antibody HRP-Anti-Human IgG was then added to the plate and kept incubated, and the wells were washed. Finally, the substrate was loaded into the wells and color development was monitored in proportion to the amount of antibodies present. The reaction was stopped by the addition of a stop solution, and the absorbance intensity was measured at 450 nm. The OD value reflects the amount of antibodies bound.

### 4.4. Detection of Anti-SARS-CoV-2 Neutralizing Antibodies by Flow Cytometry

Neutralizing antibodies against SARS-CoV-2 were analyzed using the LEGENDplex™ SARS-CoV-2 Neut Ab Assay (1-plex) w/VbP kit (BioLegend, San Diego, CA, USA, Cat #741127), according to the manufacturer’s instructions. Biolegend’s LEGENDplex™ SARS-CoV-2 Neut. Ab Assay (1-plex) immunoassay is a single-plex bead-based assay that uses the same basic principle as competitive immunoassay. Beads are conjugated with ACE2 protein. Biotinylated S1-Fc chimera and anti-human S1 recombinant antibody serve as the detection and neutralization antibodies, respectively. These were added to all wells prior to the addition of the ACE2 capture beads and competed for binding to ACE2, producing a competitive assay. Streptavidin-phycoerythrin (SA-PE) was subsequently added, which bound to the biotinylated S1-Fc detection, providing a fluorescent signal intensity in proportion to the amount of bound analyte. The concentration of a particular analyte was determined using a standard curve generated in the same assay. The competition between the detection and the standard neutralization antibody formed an inverted standard curve. Due to the nature of the assay, the standard curve had the highest signal at the lowest standard neutralization antibody concentration, and the signal decreased as the standard neutralization antibody concentration increased. Antibody concentrations were calculated by interpolation on a standard curve using logistic regression with Legendplex v.2021.07.01 software (BioLegend).

The placental transfer ratio was calculated as the ratio of neutralizing antibodies in umbilical cord blood and maternal blood. The expected transfer efficiency was >1, indicating higher umbilical cord titers compared to maternal titers at delivery [9,17].

### 4.5. Statistical Analysis

The analysis was performed using the Prism 8 statistical package (GraphPad Prism, San Diego, CA, USA).

Serological data sets were subjected to the Kolmogorov–Smirnov normality test. The difference between antibody titers in maternal and umbilical cord blood was analyzed using a Wilcoxon matched-pairs signed rank test; the correlation was analyzed using a Spearman’s correlation test. Comparisons of placental transfer efficiency by gestational age groups were analyzed using the Kruskal–Wallis test. Statistical significance was considered for *p* < 0.05.

Antibody titers and neutralization rates are presented as median and interquartile ranges (IQR); the transfer ratio is shown as mean ± standard deviation.

## Figures and Tables

**Figure 1 ijms-25-01516-f001:**
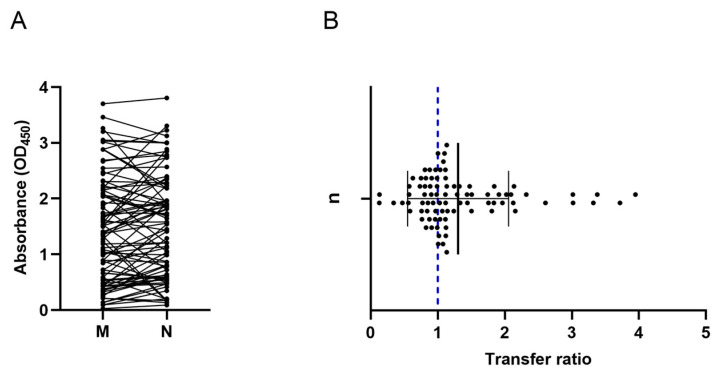
Anti-SARS-CoV-2 IgG. (**A**) Antibody titers. A correlation of maternal and neonatal antibodies was found; there are no significant differences between maternal and neonatal the concentrations. The dot plot shows the concentration of total IgG antibodies in the mother (M) and in the neonate (N); the M-N binomial is connected by a line. (**B**) Placental transfer ratio. IgG transfer was efficient (≥1.0). The dot plot shows the transfer ratio of each mother–neonate pairing. The blue dotted line represents the expected ratio. Mean ± SD is shown.

**Figure 2 ijms-25-01516-f002:**
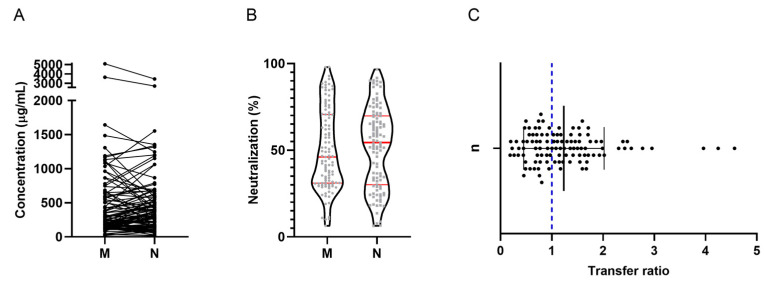
Neutralizing antibodies against SARS-CoV-2. (**A**) Antibody concentration. A correlation of maternal and neonatal antibodies was found; there are no significant differences between the concentrations per group. The dot plot shows the concentration of neutralizing antibodies in the mother (M) and in the neonate (N); the M-N binomial is connected by a line. (**B**) Percent neutralization. No differences were found between mothers and neonates. Red lines indicate median and interquartile ranges. (**C**) Placental transfer ratio. The transfer of neutralizing antibodies was efficient (≥1.0). The dot plot shows the transfer ratio of each mother–neonate pairing. The blue dotted line represents the expected rate. Mean ± SD is shown.

**Figure 3 ijms-25-01516-f003:**
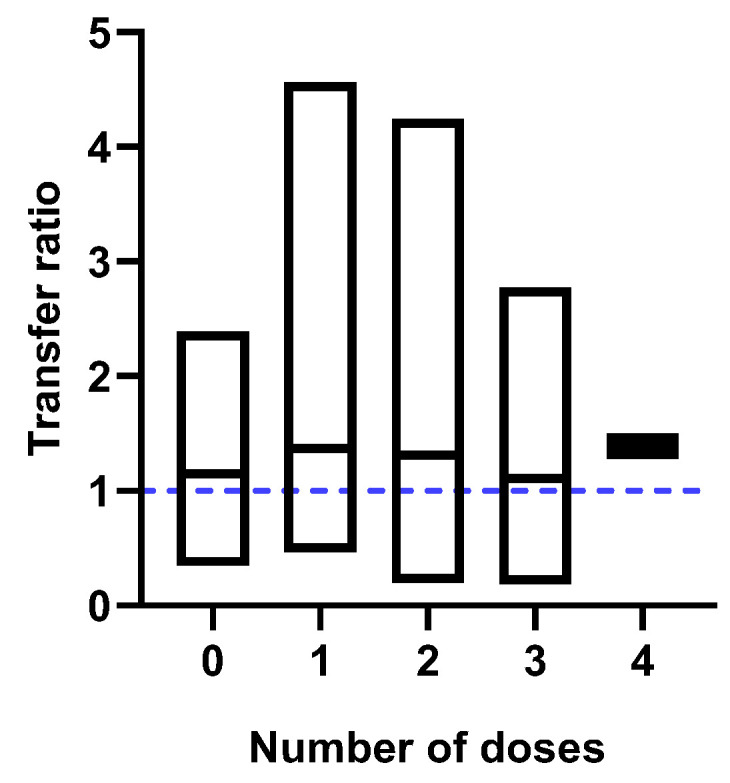
Placental transfer of nAb by number of doses administered. No significant differences were found between the number of doses received by the mother and the placental transfer efficiency. The blue dotted line represents the expected transfer ratio.

**Figure 4 ijms-25-01516-f004:**
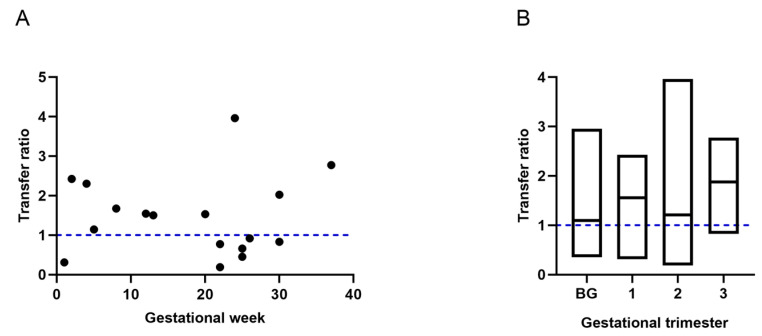
Placental transfer of nAb by gestational age at time of immunization. (**A**) Transfer efficiency per gestational week. No correlation was found between weeks of gestation and transfer efficiency. (**B**) Transfer efficiency per gestational trimester. No significant differences were found between gestational trimester and transfer efficiency. BG: before gestation. The blue dotted lines represent the expected ratio.

**Figure 5 ijms-25-01516-f005:**
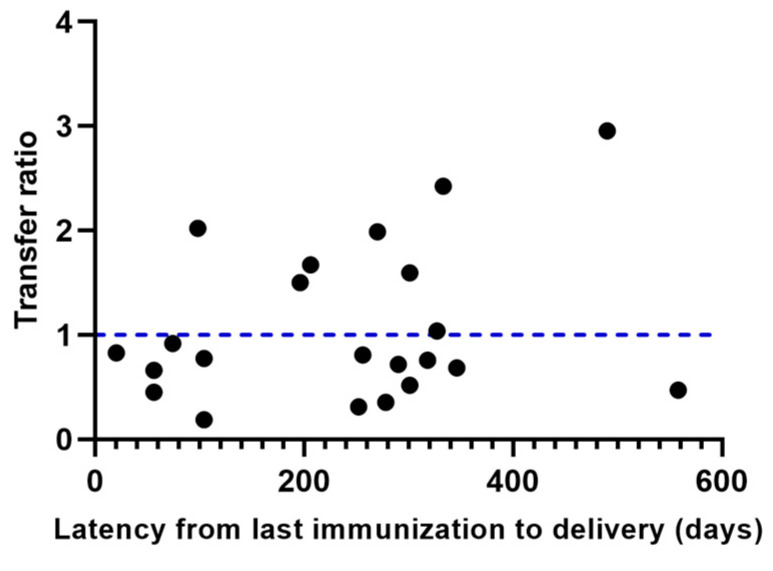
Placental transfer of nAb with respect to post-vaccination latency. No correlation was found between transfer efficiency and latency from vaccination to delivery. The blue dotted line represents the expected ratio.

**Table 1 ijms-25-01516-t001:** History of COVID-19 and vaccination in the study population.

	N (%)
Vaccination against COVID-19	
Unvaccinated	26 (22.8)
1 dose	18 (15.8)
2 doses	34 (29.8)
3 doses	33 (28.9)
4 doses	3 (2.6)

## Data Availability

Data are contained within the article.

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
