# Peer review of "Placental Transfer Efficiency of Neutralizing Antibodies on SARS-CoV-2 Vaccination before and after Pregnancy in Mexican Women"

_ijms, 2024, doi:10.3390/ijms25031516_

Round 1

Reviewer 1 Report

Comments and Suggestions for Authors

The authors brought a very inquisitive matter in realm of COVID-19 vaccination, and its neutralizing effect. This is very informative in a multidimensional platform. Besides its high enthusiasm and merit, it needs some modification as well as revision, as is felt by this reviewer, and are narrated below: 

The Title:

I would rather recommend changing the title of the manuscript to:

Placental Transfer Efficiency of Neutralizing Antibodies on SARS-CoV-2 Vaccination Before and After Pregnancy in Mexican Women 

In Abstract

Line 17: Put a comma after study.....

In Introduction

Line 28-57: To many para(s) with loosely bound list of information just tabulated. All of this information has to make a good flow of story, and at the end why this study and its importance in the field (multiple) has to be indicated. 

In Results and Discussion:

Besides being having a significant and positive correlation found in the nAb antibody concentration and responses between maternal and neonatal, majority of the other covariates are not such statistically significant in and otherwise mentioned and referred as significant as depicted 235-238 (about the vaccination platform), and 194-196. This need to be discussed and the importance that it reflects.  

Comments on the Quality of English Language

It needs few and minor changes.

Author Response

Thank you very much for taking the time to review this manuscript. Please find the detailed responses below and the corresponding revisions/corrections highlighted in the re-submitted files.

Point-by-point response to Comments and Suggestions for Authors

Comments 1: 

The tittle:

I would rather recommend changing the title of the manuscript to:
Placental Transfer Efficiency of Neutralizing Antibodies on SARS-CoV-2 Vaccination Before and After Pregnancy in Mexican Women

Response 1:

We agree with this comment. We have decided to change the title of our manuscript to the one you kindly suggested.

Comments 2: 

In Abstract:

Line 17: Put a comma after study.....

Response 2:

We've fixed the grammatical error.

Comments 3:

In Introduction:

Line 28-57: To many para(s) with loosely bound list of information just tabulated. All of this information has to make a good flow of story, and at the end why this study and its importance in the field (multiple) has to be indicated. 

Response 3:

Agree. we have, accordingly,  modified the structure and breadth of the introduction for greater fluidity.

Comments 4: 

In Results and Discussion:

Besides being having a significant and positive correlation found in the nAb antibody concentration and responses between maternal and neonatal, majority of the other covariates are not such statistically significant in and otherwise mentioned and referred as significant as depicted 235-238 (about the vaccination platform), and 194-196. This need to be discussed and the importance that it reflects. 

Response 4:

Agree. From line 256 onwards, we include more detailed data on the use of different vaccination platforms and the immune response obtained.

Reviewer 2 Report

Comments and Suggestions for Authors

Dear Authors;

Re: Manuscript ID

ijms-2782852   In this article (Original Research Paper) you aimed to evaluate the effect of corona viridea vaccination  on placental transfer efficiency of neutralizing antibodies in Mexican women. The study topic is extremely important as a current medical issue. Your manuscript is well written in general. Please double-check the sentence in Lines 24-25: "Vaccination against COVID-19 before and during pregnancy in Mexican population induces a strong and lasting neutralizing response in mothers and their newborns." It seems that a quantitative evidence is required here. Study aim must be put more clearly in the Introduction. Please re-phrase the sentence in Lines 69-72 (... only were detected in ...). In Table 1, it mentions around 23 percent of the study population were unvaccinated. Please make it clear that this group was used as a control group. Also, in other results make the control and statistical analysis more clear. For Introduction (and Discussion) regarding contemporary and advanced vaccine technologies authors are advised to consult recent publications, e.g.; PMID: 35490320  DOI: 10.2174/1567201819666220427125342 and Biomedicines 2021, 9(5), 520;  https://doi.org/10.3390/biomedicines9050520 Clinical study permissions seem to be missing. Conclusion, recommendation for future studies also missing.    

Author Response

Thank you very much for taking the time to review this manuscript. Please find the detailed responses below and the corresponding revisions/corrections highlighted in the re-submitted files.

Point-by-point response to Comments and Suggestions for Authors

Comments 1: 

 In this article (Original Research Paper) you aimed to evaluate the effect of corona viridea vaccination  on placental transfer efficiency of neutralizing antibodies in Mexican women. The study topic is extremely important as a current medical issue. Your manuscript is well written in general. Please double-check the sentence in Lines 24-25: "Vaccination against COVID-19 before and during pregnancy in Mexican population induces a strong and lasting neutralizing response in mothers and their newborns." It seems that a quantitative evidence is required here. Study aim must be put more clearly in the Introduction. Please re-phrase the sentence in Lines 69-72 (... only were detected in ...). In Table 1, it mentions around 23 percent of the study population were unvaccinated. Please make it clear that this group was used as a control group. Also, in other results make the control and statistical analysis more clear. For Introduction (and Discussion) regarding contemporary and advanced vaccine technologies authors are advised to consult recent publications, e.g.; PMID: 35490320  DOI: 10.2174/1567201819666220427125342 and Biomedicines 2021, 9(5), 520;  https://doi.org/10.3390/biomedicines9050520 Clinical study permissions seem to be missing. Conclusion, recommendation for future studies also missing. 

Response 1: 

We agree with tis comment. The sentence on lines 24 and 25 as well as that on lines 69-72 (now 87-88) were reformulated. In the results, it was clarified that unvaccinated women constituted the control group. The introduction and discussion were restructured and expanded. The objectives and conclusions were detailed accordingly.